# Integration of Web APIs and Linked Data Using SPARQL Micro-Services—Application to Biodiversity Use Cases †

**Franck Michel [1,*]**, **Catherine Faron Zucker [1]**, **Olivier Gargominy [2]** and **Fabien Gandon [1]**

[1]   I3S laboratory, University Côte d'Azur, CNRS, Inria, 930 route des Colles, 06903 Sophia Antipolis, France; faron@i3s.unice.fr (C.F.Z.); fabien.gandon@inria.fr (F.G.)

[2]   Muséum National d'Histoire Naturelle, 36 rue Geoffroy Saint-Hilaire, 75005 Paris, France; gargo@mnhn.fr

*   Correspondence: fmichel@i3s.unice.fr

†   This paper is an extended version of our conference paper: Michel F., Faron Zucker C. and Gandon F. (2018). SPARQL Micro-Services: Lightweight Integration of Web APIs and Linked Data. In Proceedings of the Linked Data on the Web (LDOW2018), Lyon, France, 23 April 2018.

**Abstract:**   In recent years, Web APIs have become a de facto standard for exchanging machine-readable data on the Web. Despite this success, however, they often fail in making resource descriptions interoperable due to the fact that they rely on proprietary vocabularies that lack formal semantics. The Linked Data principles similarly seek the massive publication of data on the Web, yet with the specific goal of ensuring semantic interoperability. Given their complementary goals, it is commonly admitted that cross-fertilization could stem from the automatic combination of Linked Data and Web APIs. Towards this goal, in this paper we leverage the micro-service architectural principles to define a *SPARQL Micro-Service* architecture, aimed at querying Web APIs using SPARQL. A SPARQL micro-service is a lightweight SPARQL endpoint that provides access to a small, resource-centric, virtual graph. In this context, we argue that full SPARQL Query expressiveness can be supported efficiently without jeopardizing servers availability. Furthermore, we demonstrate how this architecture can be used to dynamically assign dereferenceable URIs to Web API resources that do not have URIs beforehand, thus literally "bringing" Web APIs into the Web of Data. We believe that the emergence of an ecosystem of SPARQL micro-services published by independent providers would enable Linked Data-based applications to easily glean pieces of data from a wealth of distributed, scalable, and reliable services. We describe a working prototype implementation and we finally illustrate the use of SPARQL micro-services in the context of two real-life use cases related to the biodiversity domain, developed in collaboration with the French National Museum of Natural History.

**Keywords:** Web API; REST; SPARQL; micro-service; data integration; linked data; biodiversity

## 1. Introduction

With the advent of the Big Data era during the last fifteen years, many works have tackled the challenge of dealing with large volumes of data produced at a high velocity (commonly known as the first two V's of Big Data). This has led to the emergence of new types of databases (so-called NoSQL databases [1]) and new processing paradigms (e.g., MapReduce [2]). But volume and velocity are just one part of the new issues that we are given to face. The dramatic proliferation of data sources available on the Web poses an unprecedented challenge upon data integration, i.e., the techniques involved in combining heterogeneous data residing in different systems and locations, into a common, consistent view. This is where we have to tackle the third and fourth V's of Big Data: Variety

refers to data syntactic and semantic heterogeneity, while veracity addresses questions such as data interpretation, bias, uncertainty, provenance, and, ultimately, trust.

Various types of interfaces can enable access to and querying of big data sets. In particular, Web APIs have gained significant traction during the last decade, to the extent they became a de facto standard for exchanging data on the Web. They are routinely published by Web portals and data producers to enable HTTP-based, machine-processable access to their data. Let us illustrate this with a few examples. As of October 2018, the *ProgrammableWeb.com* portal was registering over 20,000 Web APIs pertaining to topics as diverse as social networks, entertainment, or finance. Many governments around the world have set up data portals accessible through Web APIs to inventory open data sets originating from the public sector [3]. In the biodiversity domain, data aggregators and Natural History museums maintain specialized Web APIs giving access to billions of records about taxonomies, occurrences, and traits of biological organisms [4].

Despite their success in ensuring a relatively uniform access to resource descriptions, Web APIs fail in making these descriptions interoperable. Indeed, they commonly rely on service-specific data models and proprietary vocabularies that are hardly connected with other ones. More importantly, these vocabularies lack a clear semantic description that is essential to address the variety and (to a lesser extend) veracity issues. The Linked Data principles [5] specifically seek to address this lack. They propose to publish data (1) in a common machine-readable format (RDF, the Resource Description Framework [6]); (2) using shared and linked vocabularies with clearly defined semantics; (3) while linking related resources throughout datasets. Links bring about the ability to consolidate and make sense of disparate datasets, thus building a uniform distributed knowledge graph. Similarly to Web APIs, large amounts of data about all sorts of topics are increasingly being published in compliance with the Linked Data principles. As of October 2018, over 25,000 Linked Data datasets were inventoried by LODAtlas (http://lodatlas.lri.fr/).

Given the complementary goals of Web APIs and Linked Data, it is commonly admitted that cross-fertilization could stem from their mutual integration. Initiatives like Google's Knowledge Graph (https://goo.gl/BqMC21) or Facebook's Open Graph (http://ogp.me/) are continuously enriching vast knowledge graphs by gathering data from these two worlds alongside other types of data sources. Strikingly however, no standard approaches have emerged so far. Several issues can be pointed out, that explain this lack:

- **Vocabularies**: Web APIs typically rely on standard representation formats such as JSON or XML, but how to invoke a Web API and interpret resource representations is usually documented in Web pages meant for human readers. By contrast, Linked Data best practices [7] advocate the use of standard protocols and common, well adopted vocabularies described in machine-readable formats. Consequently, consuming Web API data and RDF triples alike is often achieved through the development of Web API wrappers implementing bespoke vocabulary alignment.

- **Resource identifiers**: Web APIs commonly name resources using proprietary, internal identifiers. The downside is that such internal identifiers do not have any meaning beyond the scope of the Web API itself. Linked Data principles address this issue by relying on HTTP URIs to identify resources. Not only URIs are unique on the Web, but they can also establish uniform affordances of the resources in that they can be dereferenced to a description of the resource. Therefore, integrating Web APIs and Linked Data requires a mechanism to associate internal identifiers to URIs.

- **Parsimony**: Web APIs frequently consist of many different services (search by name/tag/group, organize content, interact with contents, etc.). Such that providing a Linked Data interface for all of these services may require substantial efforts, although a tiny fraction of them may fulfill the needs of most use cases. Therefore, a more parsimonious, on-demand approach may be more relevant.

- **Suitable interface**: Controversy exists with respect to the type(s) of interface(s) most suitable to query Web APIs in a way that allows for their integration with Linked Data. Each type of Linked

Data interface has its own benefits and concerns. RDF dumps allow in-house consumption but do not fit in when data change at a high pace; URI dereferencing (i.e. looking up a URI to retrieve a resource representation in a negotiated media type such as one of the RDF serialization syntaxes) provides subject-centric documents hence lacking query expressiveness. At the other end of the spectrum, SPARQL [8], the W3C recommendation to query RDF graphs, is more expressive but puts the query processing cost solely on the server, and studies suggest that allowing clients to run arbitrary SPARQL queries against public endpoints leads to availability issues [9]. Besides, on-the-fly SPARQL querying of non-RDF databases proves to be challenging, as attested by the many works on SPARQL-based access to relational [10,11] or NoSQL [12,13] databases.

In this article, we propose to harness the Semantic Web standards to enable automatic combination of disparate resource representations coming from both Linked Data interfaces and Web APIs. Our contribution is threefold. Firstly, we define the *SPARQL Micro-Service* architecture aimed at querying Web APIs using SPARQL while assigning dereferenceable URIs to Web API resources (that do not have URIs beforehand). Secondly, we suggest that leveraging the micro-service architectural principles [14] could help to design more distributed and modular Linked Data-based applications. Thirdly, we illustrate the use of SPARQL micro-services in several real-life use cases related to the biodiversity domain.

The rest of this article is organized as follows. Section 2 introduces background elements with respect to Web APIs and micro-service architectural principles. Section 3 defines the SPARQL micro-service architecture, while Sections 4 and 5 present our implementation and the experimentation we have conducted. This is then complemented in Section 6 with the description of two concrete use cases in the biodiversity area. Related works are discussed in Section 7 while the last section sums up our approach and suggests future leads.

## 2. Background

### 2.1. Web APIs

*Web APIs* are a loosely defined category of services accessed over the Web. Since the term is used in the literature to refer to somewhat different things, below we outline more precisely what it means within the context of this article.

Web APIs are invoked over the HTTP/HTTPS protocols. Unlike WSDL-based Web services, they leverage the HTTP protocol to denote actions, spawning relatively simple interfaces where service arguments are traditionally passed as regular parameters of an HTTP query string. These APIs are informally called REST interfaces although they are generally more "REST-like" than RESTful, i.e., they comply with some of the REST architectural principles [15] (stateless interaction, uniform interface) but relax others (self-contained response, resources identified with URIs, hypermedia links to navigate resource states).

Most Web APIs use a standard representation format such as XML and JSON, thereby ensuring a relatively uniform access to resource representations. These descriptions however often rely on proprietary vocabularies typically documented in Web pages meant for software developers but hardly machine-readable. This is a major difference with Linked Data best practices [7] that advocate the use of common, well adopted vocabularies whose semantics is described in machine-readable format. Some initiatives make Web APIs documentation machine-readable, such as OpenAPI (https://github.com/OAI/OpenAPI-Specification) (formerly Swagger). This description however hardly touches upon semantic concerns; it mostly consists of a syntactic description of the operations supported by the interface, and aims to automate the generation of client-side and server-side code. In this respect, OpenAPI is no different from what WSDL allowed to do. By contrast, smartAPI [16] extends OpenAPI with an extensive set of metadata, but the link with vocabularies and ontologies from the Web of Data is still to be defined.

## 2.2. Micro-Service Architectures

The term *micro-service*, also called *fine-grained SOA*, refers to an architectural style where an application builds upon a set of services that are fine-grained (they fulfill a single, well-defined function), loosely coupled and independently deployable [14]. Although no standard definition of micro-services exists so far, consensus is emerging about commonly agreed upon principles [17,18]. For instance, a proposition rewords these principles by stating that micro-services should be elastic, resilient, composable, minimal and complete (https://www.nirmata.com/2015/02/02/microservices-five-architectural-constraints/). Away from traditional monolithic applications, the micro-service architectural style improves modularity by making applications easier to develop, maintain, and test. Development teams are typically geared towards continuous refactoring, delivery and deployment of the services they are responsible for, independently of other teams and services. Instead of being constrained to use specific technologies, each team picks the programming language(s), software stacks, and other third-party technologies that it deems most appropriate for a specific service, hence giving rise to *polyglot applications*.

Micro-services have gained increasing traction during the last few years, not only from the Web companies who inspired this architecture, but also from many other companies that need to make their development and deployment processes more agile. Micro-services are often associated with lightweight container technologies such as the popular Docker (https://www.docker.com/). These technologies can underpin the quick and elastic deployment of applications by enabling on-demand scaling up or down of the micro-services.

Obviously, beyond these seemingly idealistic principles, the experience shows that micro-services have pitfalls of their own (https://www.infoq.com/news/2014/08/failing-microservices). For instance, figuring out the right functional scope of a micro-service so as to keep it minimal and complete is critical in many aspects. With "too minimal" micro-services, an application may soon consist of several hundreds of micro-services, far from the intuitive idea of "easily maintainable". Furthermore, insofar as complex services are achieved by composing multiple micro-services, "too minimal" services tend to entail complex dependency networks that, again, cause maintainability issues.

Still, despite these downsides, we argue that leveraging micro-service principles may foster the design of Linked Data-based applications in a more distributed and modular way, by building on independent and lightweight services. Such services may typically be URI dereferencing services, SPARQL endpoints, triple stores, Linked Data Platform [19] components etc. The SPARQL micro-service architecture is a proposition designed with this mindset.

## 3. The SPARQL Micro-Service Architecture

### 3.1. Definition

A SPARQL micro-service $S_\mu$ is a wrapper of a service $S_w$ of a Web API, that allows querying $S_w$ using SPARQL. $S_\mu$ behaves as a regular SPARQL endpoint insofar as it supports the SPARQL Query Language [8] (including all query forms: SELECT, ASK, CONSTRUCT, DESCRIBE) and the SPARQL Protocol [20]. Accordingly, a client invokes $S_\mu$ over HTTP/HTTPS by passing a SPARQL query $Q$ and optional default and named graph URIs (arguments *default-graph-uri* and *named-graph-uri*).

Additionally, $S_\mu$ expects a possibly empty set $Arg_w$ of arguments that are specific to the service being wrapped. In turn, $S_\mu$ invokes $S_w$ with the arguments in $Arg_w$, and produces a SPARQL result set that matches $Q$. Figure 1 illustrates this architecture.

The semantics of SPARQL micro-services differs from that of standard SPARQL endpoints in that a SPARQL micro-service is a configurable SPARQL endpoint whose arguments ($Arg_w$) delineate the virtual graph being queried. Formally, each pair ($S_\mu$, $Arg_w$) is a standard SPARQL endpoint. As we see it, a SPARQL micro-service provides access to a small, resource-centric graph that corresponds to a small fragment of the whole dataset served by the Web API. This graph is shaped at runtime by (i) the

Web API service being wrapped; (ii) the set $Arg_w$ of arguments passed to micro-service $S_\mu$; and (iii) the types of RDF triples that $S_\mu$ is designed to produce.

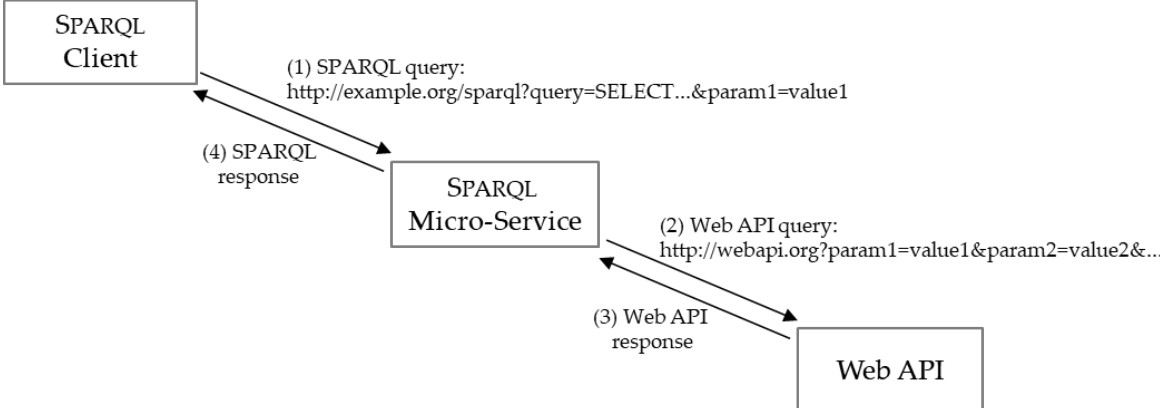

**Figure 1.** Architecture of a SPARQL micro-service with one argument `param1`. In this example, the argument is passed as a parameter on the HTTP query string.

How $Arg_w$ is passed to micro-service $S_\mu$ is implementation-dependent. The implementation we describe in Section 4 passes the arguments as parameters of the HTTP query string, and we discuss several possible alternatives in Section 4.4. Likewise, how the Web API response is transformed into a SPARQL result set matching $Q$ is implementation-dependent. Various methods may be used at this point, involving e.g., different flavors of mapping, reasoning or rule processing. In the context of JSON-based Web APIs, our implementation applies a JSON-LD profile to the Web API response and executes an optional SPARQL query to yield additional triples.

With regards to the micro-service architectural principles, we think of SPARQL micro-services as lightweight, independent software units being developed along with the arising of needs. A micro-service development team focuses on one Web API at a time, defines how to wrap the services of interest for clearly identified use cases, tests and deploys the micro-services. Using a container infrastructure like Docker, each micro-service is deployed within a dedicated container that can scale up or down automatically in response to changes in the workload.

**Example 1.** *Let $S_w$ be a service of Flickr's Web API (Flickr is a photography-related social network; its API is described at https://www.flickr.com/services/api/), that retrieves photos posted in a specific Flickr group and having a specific tag (https://www.flickr.com/services/api/flickr.groups.pools.getPhotos.html). $Arg_w$ (the arguments of $S_w$) comprises two arguments, group_id and tags. Let $S_\mu$ be a SPARQL micro-service that wraps $S_w$ and takes its arguments as parameters on the HTTP query string.*

*Let us assume a client wants to retrieve the URLs of photos of dolphins, posted to the Encyclopedia of Life Images group (https://www.flickr.com/groups/806927@N20) wherein photos are tagged with the species scientific name formatted as "taxonomy:binomial=\<scientific name\>". To do so, the client executes the SPARQL query below against any SPARQL 1.1 endpoint, in which the SERVICE clause invokes $S_\mu$ while passing the group_id and tags arguments:*

```
PREFIX s: <http://schema.org/>
SELECT * WHERE {
SERVICE <http://example.org/flickr/getPhotosByGroupByTag? \
group_id=806927@N20&tags=taxonomy:binomial=Delphinus+delphis>
{ SELECT ?img WHERE { ?photo s:image ?img. } }
}
```

*$S_\mu$ invokes Flickr's Web API service $S_w$ with the arguments in $Arg_w$, possibly in addition to technical arguments required by the Web API. Listing 1 shows the invocation URL (top) and sketches a snippet of the response (bottom). $S_\mu$ translates this response into an RDF graph G such as the one depicted in Listing 2.*

*Remember that how this translation is carried out is implementation-dependent. Finally, $S_\mu$ evaluates the client's SPARQL query against G and returns the response in one of the media types supported by the SPARQL client (following a regular content negotiation [21]).*

**Listing 1.** Top: Invocation of a Flickr's Web API service (technical parameters were left over for clarity). Bottom: Snippet of the JSON response.

```
https://api.flickr.com/services/rest/?
  method=flickr.groups.pools.getPhotos&format=json&per_page=500&
  group_id=35034350743@N01&tags=taxonomy:binomial=Delphinus+delphis

{ "photos": {
    "page": 1, "pages": "15",
    "photo": [
      { "id": "31173091406", "title": "Delphinus delphis 5 (13-7-16 San Diego)",
        "owner": "10770266@N04",  "ownername": "Barbol",
        "secret": "8c7437e970", "server": "5718", "farm": 6 }
    ]
}}
```

**Listing 2.** Example of RDF graph representing the Web API response in Listing 1, in the Turtle syntax [22].

```
PREFIX s: <http://schema.org/>

<http://example.org/ld/flickr/photo/31173091626>
  a s:Photograph ;
  s:name "Delphinus delphis 5 (13-7-16 San Diego)";
  s:image <https://farm6.staticflickr.com/5718/31173091626_88c410c3f2_z.jpg>;
  s:mainEntityOfPage <https://flickr.com/photos/10770266@N04/31173091626>;
  s:author [
    s:name "Barbol";
    s:url <https://flickr.com/photos/10770266@N04>
  ].
```

## 3.2. Assigning URIs to Web API Resources

Web APIs usually identify the resources they manage with internal, proprietary identifiers. For instance, string "31173091406" identifies the photographic resource represented in Listing 1. But beyond the scope of Flickr's Web API, this identifier is totally meaningless. As a first approach, the URL of this photo's Web page could serve as a URI. This practice is however discouraged as it tends to mix up the resource (a photographic work in this case) with an HTML representation thereof.

Therefore, bridging Web APIs and Linked Data not only requires to enable SPARQL querying of Web APIs, but also to dynamically create URIs that identify Web API resources. Furthermore, according to Linked Data best practices [7], it should be possible to look up these URIs in order to retrieve a description of the resources in a negotiated media type. Conventionally, dereferencing a URI returns a set of RDF triples where the URI is either in the subject or object position. This is typically achieved through a CONSTRUCT or DESCRIBE SPARQL query.

SPARQL micro-services can be used to implement URI dereferencing in a straightforward manner. Once a domain name and URI scheme are chosen, a Web server can be set up to deal with this URI scheme. When the Web server receives a look-up query for a URI that matches the scheme, it generates an invocation of the relevant SPARQL micro-service. Technically, the Web server acts as a reverse proxy: It queries the SPARQL micro-service and transparently proxies the response back to the client. Figure 2 sketches this architecture.

Hence, by smartly designing SPARQL micro-services, we can build a consistent ecosystem where some micro-services respond to SPARQL queries by assigning URIs to Web API resources, while some micro-services (possibly the same) are able to dereference these URIs to an RDF content.

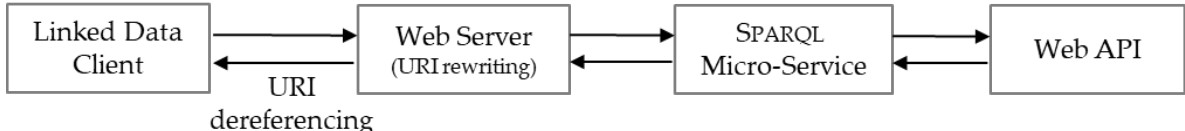

**Figure 2.** Using a SPARQL micro-service to dereference URIs to RDF content.

**Example 2.** *Let us assume that the getPhotoById SPARQL micro-service retrieves photos by their Flickr identifier (argument photo_id). When the Web server receives a look-up query for URI "http://example.org/ ld/flickr/photo/38427227466", it invokes the getPhotoById service with the following inputs: (i) The photo identifier passed as argument photo_id; (ii) a SPARQL query to retrieve a graph representing the resource, typically a DESCRIBE query on the URI being looked up; (iii) the Accept HTTP header from the look-up query, to enable end-to-end content negotiation. An example of the URL generated by the Web server in response to this URI look-up is shown below:*

```
http://example.org/flickr/getPhotosById?
photo_id=38427227466&
query=DESCRIBE\%20\%3Chttp\%3A\%2F\%2Fexample.org\%2Fld \
\%2Fflickr\%2Fphoto\%2F38427227466\%3E
```

*The Web server finally proxies the response back to the client in one of the negotiated media types.*

*If the SPARQL query is more verbose that in the example above (e.g., a large CONSTRUCT query), the Web server rewriting configuration may become cumbersome and more difficult to maintain. The implementation described in Section 4 mitigates this issue by allowing each micro-service to provide a CONSTRUCT query that shall be used to answer a URI dereferencing query.*

### 3.3. Analysis of SPARQL Micro-Services with the Linked Data Fragments Framework

There exist several types of interfaces to interact with and query Linked Data. The *Linked Data Fragments* [23] framework was designed to enable the analysis and comparison of such Linked Data interfaces. Within this framework, a query response returned by a Linked Data interface is called a *Linked Data Fragment* (LDF). Different types of LDF can be sorted out according to their query granularity and the way processing workload is balanced between a server and a client. Common LDF types are depicted on Figure 3: on the left hand-side, querying consists of a mere download operation and clients bear the full cost of evaluating queries against RDF dumps; on the right hand-side, SPARQL enables expressing specific queries but endpoints fully bear the evaluation cost. Between these two extremes lies a spectrum of intermediate approaches. A Linked Data document results from a URI dereferencing. A *Triple Pattern Fragment* [23] (TPF) results from evaluating a single triple pattern against an RDF dataset. The TPF approach succeeds in balancing the query evaluation cost between the client and the server, thereby ensuring better server availability than full-fledged SPARQL endpoints, at the cost of a loss of efficiency. SaGe [24] mitigates this issue with a modification of the SPARQL protocol. A SaGe server can interrupt the processing of a SPARQL query and send back to the client the current results alongside the information needed by the server to carry on processing during a subsequent invocation.

We see SPARQL micro-services as an alternative Linked Data Fragment interface. A SPARQL micro-service $S_\mu$ provides access to a virtual graph that corresponds to a fragment of the dataset served by the Web API. By designing it carefully, we can ensure that $S_\mu$ will yield small (typically resource-centric) virtual graphs. In this context, virtually any query can be processed at low cost. In other words, full SPARQL Query expressiveness can be supported without jeopardizing the service availability. Consequently, in Figure 3, SPARQL micro-service results stand nearby SPARQL results

since SPARQL micro-services allow requests as specific (expressive) as SPARQL, and to the right of SaGe results since they do not put any additional processing cost on the client.

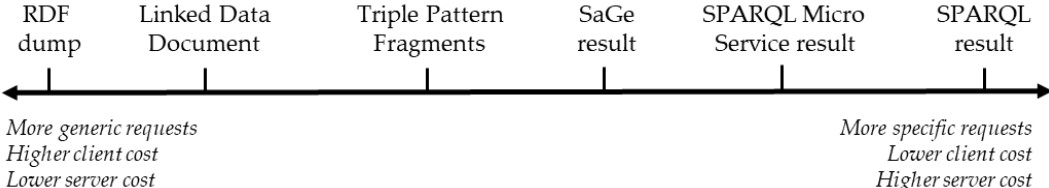

**Figure 3.** Granularity of different types of *Linked Data Fragments*.

## 4. Implementation

To evaluate the architecture proposed in Section 3, we have developed a prototype implementation written in the PHP language and available on GitHub (https://github.com/frmichel/sparql-micro-service/tree/0.1.0/) under the Apache 2.0 license. This prototype targets Web APIs able to return a JSON response. This pragmatic choice stems from the fact that most modern Web APIs support JSON. Nevertheless, it would be easy to extend the prototype with respect to other formats such as XML.

Also, the prototype requires that the $Arg_w$ set of arguments of a micro-service be passed as parameters on the HTTP query string (as exemplified in Section 3.1). Thus, while the URL of a regular SPARQL endpoint is of the form "http://example.org/sparql", in our implementation a SPARQL micro-service is invoked with additional parameters, like "http://example.org/sparql?param1=value1". Alternatives to this choice are discussed in Section 4.4.

### 4.1. Processing SPARQL Queries

A SPARQL micro-service evaluates a query against an RDF graph built at run-time from data obtained from the Web API. The steps involved in this evaluation are depicted in Figure 4, and we describe them further on in Algorithm 1 (the algorithm's step numbers match those in Figure 1).

---

**Algorithm 1** Evaluation of a SPARQL query by a SPARQL micro-service $S_\mu$.

---

1. $S_\mu$ receives a SPARQL query $Q$ along with the arguments in $Arg_w$.
2. $S_\mu$ invokes the Web API service with the arguments in $Arg_w$ along with other parameters possibly required by the Web API.
3. $S_\mu$ translates the Web API response (formatted in JSON) into an RDF graph: first, it applies a JSON-LD profile to the response and loads the resulting graph $G$ into a triple store; then, it runs an optional SPARQL INSERT query that enriches $G$ with additional triples.
4. $S_\mu$ evaluates $Q$ against $G$ and returns the result to the client.

---

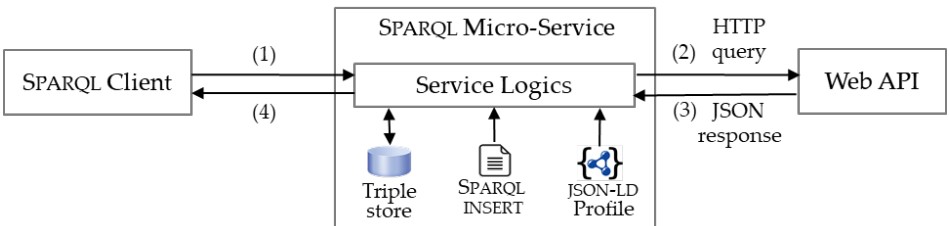

**Figure 4.** Prototype implementation of SPARQL micro-services. In step 3, a JSON-LD profile first translates the Web API JSON response into a temporary RDF graph $G$ stored in the local triple store. An optional INSERT query then augments $G$ with triples that JSON-LD cannot yield. Lastly, the client's query is evaluated against $G$.

**Example 3.** *To illustrate this algorithm, we follow up on the example introduced in Section 3.1. Steps 1 and 2 are as exemplified in Listing 1. In step 3, let us assume that the getPhotosByGroupByTag micro-service provides*

the JSON-LD profile below, that turns each JSON field name into an ad hoc RDF predicate within name space "http://sms.i3s.unice.fr/schema/" (note that any arbitrary complex profile may be used at this point):

```
{ "@context": {
"@vocab": "http://sms.i3s.unice.fr/schema/"
}}
```

The resulting graph G is stored into the local triple store. It consists of various proprietary and somehow technical predicates. For instance, the servers and farms identifiers are of little interest for a Linked Data representation, but the API documentation describes how to reconstruct the URLs of photos and users pages from these fields. This however involves the concatenation of several fields, that JSON-LD is typically incapable of describing. Therefore, the micro-service provides an INSERT query shown in Listing 3, that augments G with triples based on the Schema.org vocabulary. Listing 4 shows the graph obtained after completing both steps. The two blank nodes correspond to the application of the JSON-LD profile, while the other triples result of executing the INSERT query.

Finally, the client's SPARQL query is evaluated against G and the response is returned to the client. In the example, it consists of a solution binding for variable ?img, presented below in the SPARQL Results JSON format [25]:

```
{ "head": {
"vars": [ "img" ] },
"results": {
"bindings": [
{ "img": {
"type": "uri",
"value":
"https://farm6.staticflickr.com/5718/31173091626_88c410c3f2_z.jpg" }
}
] } }
```

**Listing 3.** Insertion of RDF triples based on the Schema.org vocabulary.

```
PREFIX api: <http://sms.i3s.unice.fr/schema/>
PREFIX s:   <http://schema.org/>

INSERT {
  ?photoUri a s:Photograph;
    s:name ?title;
    s:image ?img
    s:mainEntityOfPage ?page;
    s:author [ s:name ?authorName; s:url ?authorUrl ].
}
WHERE {
  ?photo api:id ?id; api:secret ?secret;
    api:server ?server; api:farm ?farm;
    api:title ?title; api:owner ?owner; api:ownername ?authorName.

  BIND (IRI(concat("http://example.org/ld/flickr/photo/", ?id)) AS ?photoUri)
  BIND (IRI(concat("https://flickr.com/photos/", ?owner, "/", ?id)) AS ?page)
  BIND (IRI(concat("https://flickr.com/photos/", ?owner)) AS ?authorUrl)
  BIND (IRI(concat("https://farm", ?farm, ".staticflickr.com/",
                   ?server, "/", ?id, "_", ?secret, "_z.jpg")) AS ?img)
}
```

**Listing 4.** RDF graph produced from the Web API response in Listing 1 by applying successively a JSON-LD profile and the INSERT query in Listing 3.

```
PREFIX api: <http://sms.i3s.unice.fr/schema/>
PREFIX s:   <http://schema.org/>

_:b123
  api:page 1; api:pages 15;
  api:photo _:b456.

_:b456
  api:id "31173091406"; api:title "Delphinus delphis 5 (13-7-16 San Diego)";
  api:owner "10770266@N04"; api:ownername "BioDivLibrary";
  api:secret "8c7437e970"; api:server "5718"; api:farm 6.

<http://example.org/ld/flickr/photo/31173091626>
  a s:Photograph;
  s:name "Delphinus delphis 5 (13-7-16 San Diego)";
  s:image <https://farm6.staticflickr.com/5718/31173091626_88c410c3f2_z.jpg>;
  s:mainEntityOfPage <https://flickr.com/photos/10770266@N04/31173091626>;
  s:author [
    s:name "Barbol";
    s:url <https://flickr.com/photos/10770266@N04>
  ].
```

### 4.2. URIs Dereferencing

The URIs dereferencing solution portrayed in Section 3.2 relies on the Web server to rewrite a URI into an SPARQL micro-service invocation. In particular, this invocation must provide the SPARQL micro-service with a SPARQL query. While this can easily be maintained when the query consists of a simple "DESCRIBE <uri>", richer CONSTRUCT queries may be required to accommodate different use cases. Maintaining such rewriting rules in the Web server configuarion may become cumbersome and error-prone.

To cope with this issue, our prototype proposes a more flexible alternative: each micro-service may provide a CONSTRUCT query that shall be used to answer a URI look-up query. In addition, a micro-service checks an optional parameter *query_mode* whose values may be *sparql* (the default value) or *ld*. The latter instructs the micro-service to ignore any SPARQL query passed as parameter and instead return the result of the CONSTRUCT query.

For instance, when a client looks up URI "http://example.org/ld/flickr/photo/38427227466", the Web server rewrites this into an invocation to the appropriate micro-service, for instance:

```
http://example.org/flickr/getPhotosById?photo_id=38427227466&query_mode=ld
```

### 4.3. Deployment

***Deploying a new SPARQL micro-service***. The deployment of a SPARQL micro-service in our prototype simply consists of provisioning two mandatory files and two optional files:

- *config.ini*: The micro-service main configuration file declares the arguments expected by the micro-service alongside the Web API invocation query string;
- *profile.jsonld*: The JSON-LD profile is used to translate a response from the Web API into an RDF graph;
- *insert.sparql*: This optional file provides an INSERT query meant to yield additional triples typically based on common vocabularies and ontologies;
- *construct.sparql*: This optional file provides a CONSTRUCT query meant to produce the response to URI look-up queries.

In our experience, deploying a new SPARQL micro-service is a matter of just a few hours. The most time-consuming tasks lie in reading the Web API documentation and deciding on the mapping towards domain vocabularies. Thence, a developer defines the API query string and the arguments passed to the SPARQL micro-service. Lastly, she writes the JSON-LD profile and the optional INSERT and CONSTRUCT queries that carry out the mappings.

In case a Web API requires specific actions to be taken, such as following an authentication workflow or issuing an intermediate query, the developer can customize a simple provided script (https://github.com/frmichel/sparql-micro-service/tree/0.1.0/src/sparqlms/manual_config_example/service.php), allowing for more flexibility.

*Caching Strategy*. Querying Web APIs typically takes in the order of 0.5 to 1 s, as suggested by the measures we report in Section 5. Therefore, whenever possible, a caching strategy should be defined with respect to the expected performance. There typically exist many syntactical variants of the same SPARQL query, hence classic HTTP cache servers set up between SPARQL clients and servers fail to reach efficient cache reuse. By contrast, Web API queries allow a lesser syntactical variability. Thus, in the context of SPARQL micro-services, enforcing a cache strategy on the Web API side should ensure better cache reuse.

Our prototype enforces a simple cache strategy where each Web API response is stored in a cache database, indexed with a hash of the Web API query, and annotated with an expiration period that can be configured on a per-micro-service basis. Our live deployment utilizes the MongoDB document store as a cache database, yet using any other database may be achieved with little changes. Besides, future developments may consist in adapting the caching strategy based on the data expiration information provided by some Web APIs (such as the *Expires*, *Cache-Control*, and/or *Last-Modified* HTTP headers).

*Docker Deployment*. In addition to the code available on GitHub, we have created a Docker image published on Docker Hub (https://hub.docker.com/u/frmichel/): It provides an Apache Web server configured with the SPARQL micro-services described in Section 5. Note that, for the sake of simplicity, we have defined a single image hosting several micro-services. Nevertheless, more in line with common micro-service practices, it would make sense to define one image per service, enabling the independent deployment of each service. As instructed in the GitHub README, a single command is sufficient to deploy this image on a Docker server. The deployment comes along with two more images: the Corese-KGRAM in-memory triple store [26] used to store temporary RDF graphs, and the MongoDB database used as a cache for previously executed Web API queries.

*4.4. Discussion*

A design choice of our implementation is to pass the arguments of $Arg_w$ to $S_\mu$ as parameters of the HTTP query string. Arguably, other solutions may be adopted, notably to pass the arguments as RDF terms within the SPARQL graph pattern. Below we discuss the respective benefits and drawbacks of some alternatives we identified.

A first alternative consists in defining one predicate for each argument, e.g., `api:group_id` and `api:tags` in the example below:

```
PREFIX api: <http://sms.i3s.unice.fr/schema/>
PREFIX s:   <http://schema.org/>
SELECT ?img WHERE {
SERVICE <http://example.org/flickr/getPhotosByGroupByTag>
{ ?photo s:image ?img;
api:group_id "806927@N20";
api:tags "taxonomy:binomial=Delphinus+delphis".
}
}
```

At a first sight, making the arguments explicit in the graph pattern can seem compelling in several respects. (i) They can be used in other parts of the graph pattern, for instance to invoke other SPARQL micro-services; (ii) Reusing terms from well-adopted vocabularies attaches a clearly-defined semantics to each argument; (iii) A SPARQL micro-service defined this way is a standard SPARQL endpoint since there is no more variable part in the service endpoint URL.

Several concerns can be pointed out however. (i) We seek a solution wherein only terms of well-adopted vocabularies would be exposed. Whether such terms exist would be very dependent on the nature of the arguments. For instance, the `tags` argument is likely to have equivalent terms in third-party vocabularies such as https://schema.org/keywords. But the `group_id` argument is more tightly related to its use within Flickr, and we may have to coin a new term like `api:group_id` in the example above. This approach bears the risk that each SPARQL micro-service would be defined along with its own bespoke terms, thus missing the vocabulary reuse objective; (ii) Furthermore, the `group_id` and `tags` arguments are meaningful for the end user. But some services may require more technical arguments that we typically do not want to define as ontological terms; (iii) This solution also questions the nature of the subject to which the arguments are associated. Again, in this specific example, declaring the `group_id` and `tags` as properties of the photographic resource `?photo` makes sense, but this could be inappropriate with more internal or technical service parameters; (iv) finally, providing the arguments as terms of the SPARQL graph pattern requires a mechanism to identify these specific terms as the service inputs.

Some of these issues can be solved by associating the arguments to a separate resource depicting the service itself. This is exemplified in the second alternative that, furthermore, defines a vocabulary to pass the arguments in a uniform manner. Note that existing vocabularies may be tapped for that matter, such as Hydra [27] or the Schema.org actions vocabulary (Schema.org actions: https://schema.org/docs/actions.html). In the example below, additional triple patterns define an instance of the hypothetical `api:Service` class, that takes arguments declared with the `api:param` predicate.

```
SELECT ?img WHERE {
SERVICE <http://example.org/flickr/getPhotosByGroupByTag>
{ ?photo s:image ?img.

[] a api:Service;
api:param [ api:name "group_id"; api:value "806927@N20" ];
api:param [ api:name "tags";    api:value ?tag ].
}
}
```

A slight variation could state that the service URL itself is an instance of `api:Service`; the arguments would then configure an execution of this service with predicate `api:execution`, e.g.,:

```
SELECT ?img WHERE {
SERVICE <http://example.org/flickr/getPhotosByGroupByTag>
{ ?photo s:image ?img.

<http://example.org/flickr/getPhotosByGroupByTag> a api:Service;
api:execution [
api:param [ api:name "group_id"; api:value "806927@N20" ];
api:param [ api:name "tags";    api:value ?tag ].
].
}
}
```

While these alternatives avoid defining new predicates for each micro-service, the additional triples bear a somewhat artificial semantics: they provide the service with information as to how to process the other parts of the graph pattern, but they do not actually refer to nor describe the photographic resources that the graph pattern aims to match.

In a third alternative, the service arguments are passed as SPARQL variables with pre-defined names, e.g., `?group_id` and `?tags` in the example below:

```
SELECT ?img WHERE {
SERVICE <http://example.org/flickr/getPhotosByGroupByTag>
{ ?photo s:image ?img.

BIND("806927@N20" AS ?group_id)
BIND("taxonomy:binomial=Delphinus+delphis" AS ?tags)
}
}
```

Similarly to the previous alternative, variables `?group_id` and `?tags` are somewhat artificial insofar as they provide the service with information as to how to process the other parts of the graph pattern.

The solution proposed in this article is a trade-off meant to satisfy certain goals. Above, we have discussed some alternative solutions, and others may probably be figured out. We believe that further discussions should be engaged to assess the benefits and concerns of these alternatives with respect to the contexts and goals.

## 5. Experimentation

To evaluate the effectiveness and efficiency of our approach, we conducted a series of tests related to the biodiversity domain. We wrote several SPARQL micro-services to wrap Web APIs providing various kinds of information about living species (all the services mentioned in this section are available on the project's GitHub repository):

1. *bhl/getArticlesByTaxon* retrieves scientific articles mentioning a given species name from the Biodiversity Heritage Library (BHL) (http://biodiversitylibrary.org/).
2. *flickr/getPhotosByGroupByTag*, already described in Section 3, is used to search the *Encyclopedia of Life Images* Flickr group (https://www.flickr.com/groups/806927@N20) for photos of a given species. Photos of this group are tagged with the scientific name of the species they represent, formatted as "taxonomy:binomial=<scientific name>".
3. *macaulaylibrary/getAudioById* retrieves audio recordings for a given species identifier from the Macaulay Library (https://www.macaulaylibrary.org/), a scientific media archive related to birds, amphibians, fishes, and mammals.
4. *musicbrainz/getSongByName* searches the MusicBrainz music information encyclopedia (https://musicbrainz.org/) for music tunes whose title match a given name with a minimum confidence of 90%.

*Test Environment*. The tests were performed on a CentOS Linux 7.5 server running on a virtual machine equipped with 4 CPU cores (3.50 GHz) and 32 GB of RAM. The SPARQL micro-services were served by an Apache 2.4 Web server. The server also hosted two additional services required for the experimentation: The Corese-KGRAM RDF triple store used to store temporary graphs and the MongoDB document store used as a cache database.

### 5.1. Performance of Individual SPARQL Micro-Services

Each SPARQL micro-service was individually invoked using the *curl* command. The point was always the same: retrieve information about the common dolphin species (Delphinus delphis). The SPARQL query passed to the service was simply meant to retrieve all the triples generated during the execution: `CONSTRUCT WHERE {?s ?p ?o}`. During each invocation, a metrology service implemented in the SPARQL micro-service carried out two measures: (i) The time it takes to execute solely the Web API query; and (ii) the time it takes to complete the SPARQL micro-service invocation (the latter encompasses the earlier).

Table 1 reports the execution time for each SPARQL micro-service with no cache, averaged over ten runs. Figure 5 depicts the same measures in a more synthetic way. Column "*Triples produced*" of Table 1 gives the number of triples generated by applying subsequently the JSON-LD profile to the API response and the optional INSERT query (to spawn additional triples). It is interesting to notice that the overhead imposed by the SPARQL micro-service (in orange in Figure 5) always remains low, ranging from 25 ms to 80 ms. It accounts for 4.11% of the total time for Macaulay library's API to 25.5% for MusicBrainz's API which is by far the fastest of the four APIs.

**Table 1.** Cache-less query execution time (in seconds) against a SPARQL micro-service. The last column is the overhead (in %) imposed by the SPARQL micro-service compared to a direct Web API query.

| Web API | Triples Produced | SPARQL μ-service Exec. Time | Web API Exec. Time | Overhead | Overhead (Percentage) |
|---|---|---|---|---|---|
| Biodiversity Heritage Lib. | 1161 | $0.950 \pm 0.049$ | $0.870 \pm 0.048$ | $0.080 \pm 0.004$ | $9.24\% \pm 0.44$ |
| Flickr | 336 | $0.607 \pm 0.039$ | $0.564 \pm 0.039$ | $0.044 \pm 0.008$ | $7.74\% \pm 1.45$ |
| Macaulay Library | 87 | $0.642 \pm 0.050$ | $0.617 \pm 0.005$ | $0.025 \pm 0.004$ | $4.11\% \pm 0.07$ |
| MusicBrainz | 1160 | $0.391 \pm 0.011$ | $0.312 \pm 0.009$ | $0.079 \pm 0.004$ | $25.5\% \pm 1.22$ |

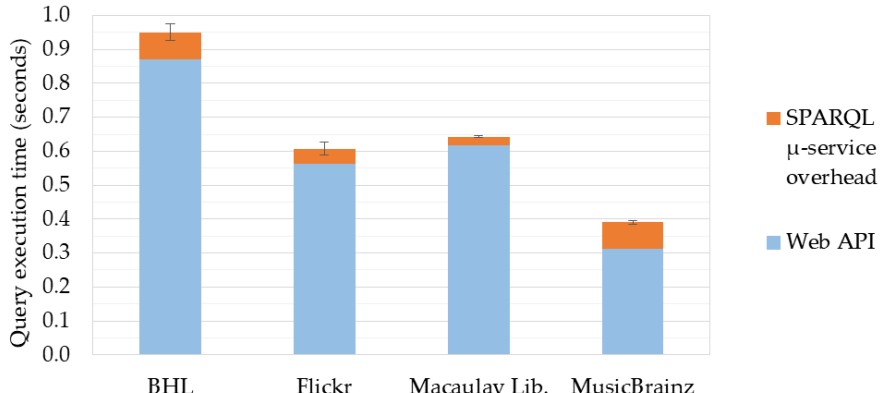

**Figure 5.** Overhead imposed by a SPARQL micro-service invocation as compared to a direct Web API invocation.

We then performed a second run of the same queries using the cache database. The cache expiration duration was set to several days and a warm-up run was executed to ensure systematic cache hits. Table 2 reports the execution time for each SPARQL micro-service without and with cache, and the last column gives the percentage of time reduction obtained when using the cache. Not surprisingly, the reduction is substantial since there is no more invocation of the Web APIs, only a document fetch operation from the local MongoDB database. The reduction ranges from 31.7% for MusicBrainz' API to 91.5% for Macaulay Library's API. Figure 6 depicts the same measures where the orange part represents the SPARQL micro-service execution with a cache database, while the blue part represents the additional time taken by the cacheless execution.

**Table 2.** Query execution time (in seconds) against a SPARQL micro-service without/with cache. The last column is the percentage of time reduction obtained when using the cache.

| Web API | SPARQL μ-Service Exec. Time without Cache | SPARQL μ-Service Exec. Time with Cache | Reduction (Percentage) |
|---|---|---|---|
| Biodiversity Heritage Lib. | $0.950 \pm 0.048$ | $0.264 \pm 0.015$ | 72.3% |
| Flickr | $0.607 \pm 0.039$ | $0.080 \pm 0.012$ | 86.9% |
| Macaulay Library | $0.642 \pm 0.050$ | $0.547 \pm 0.010$ | 91.5% |
| MusicBrainz | $0.391 \pm 0.011$ | $0.267 \pm 0.027$ | 31.7% |

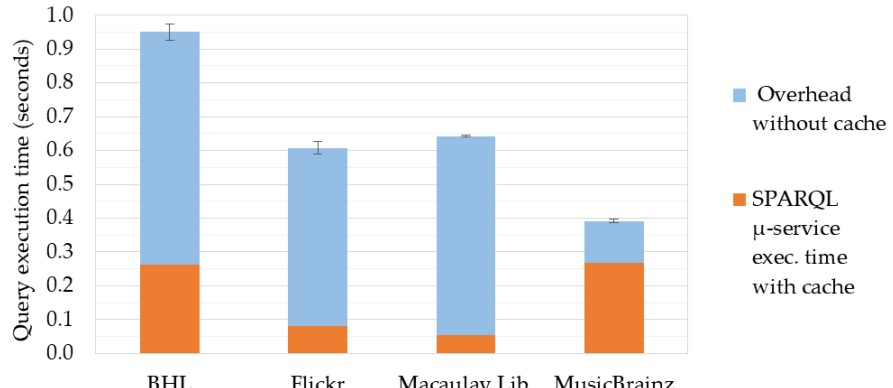

**Figure 6.** Impact of a cache database on the query execution time (in seconds) against a SPARQL micro-service.

### 5.2. Performance When Invoking Multiple SPARQL Micro-Services

After comparing the performance of individual SPARQL micro-services with the performance of the Web APIs they wrap, we designed more realistic tests aimed to gather information simultaneously from several SPARQL micro-services. In particular, we wrote a SPARQL query that invokes the four SPARQL micro-services introduced above within dedicated SERVICE clauses. It retrieves 7 articles from BHL, 15 photos from Flickr, 27 audio recordings from the Macaulay Library, and 1 music tune from MusicBrainz. We evaluated the query against two SPARQL engines: the standalone Corese-KGRAM v4.0.1 triple store that embeds a SPARQL client and a SPARQL server (we used a separate instance from that used to store temporary graphs, installed on a different machine), and the Virtuoso OS Edition v7.20 server (Virtuoso OS Edition: http://vos.openlinksw.com/owiki/wiki/VOS/).

We tested two versions of the query: in $Q_1$ (Listing 5), the micro-service endpoint URLs are provided statically, while in $Q_2$ (Listing 6), they are built dynamically and bound to SPARQL variables. Note that, concerning the Macaulay Library, queries $Q_1$ and $Q_2$ use the *macaulaylibrary/getAudioByTaxon* service that is very similar to *macaulaylibrary/getAudioById* described in Section 5.1, but retrieves audio recordings by taxon scientific name, not by identifier. The aim is to have all services being invoked consistently with the same parameter: A taxon name.

Corese-KGRAM could properly evaluate the two queries. Conversely, Virtuoso could evaluate $Q_1$ (static service URLs) but failed on $Q_2$ due to the variable service endpoints. Note that the support of this feature is not in the normative part of the SPARQL 1.1 Federated Query recommendation [28]. An implementation is free to support it or not, but the semantics is not formally defined.

Table 3 compares the execution times of query $Q_1$ on Corese-KGRAM and Virtuoso, with or without cache, averaged over ten runs. Strikingly, it highlights the difference between the two SPARQL engine strategies: The time it takes for Virtuoso to complete the query execution is bigger than that of Corese-KGRAM by a factor 53 with cache (1 min 30 s vs. 1.68 s) and a factor 112 without cache (6 min 51 s vs. 3.66 s). A detailed analysis of the query log revealed that Corese-KGRAM performed 4 invocations (one for each SERVICE clause as one would expect) whereas Virtuoso performed 424 invocations. In chronological order the MusicBrainz micro-service was invoked once, Macaulay Library once, Flickr 27 times and BHL 395 times.We discussed this issue with Virtuoso's developers (Virtuoso GitHub issue: https://github.com/openlink/virtuoso-opensource/issues/724) who could not bring a clear explanation so far. Nevertheless, further tests with two or three SERVICE clauses suggested that each SERVICE clause is invoked one time for each solution retrieved from previously evaluated SERVICE clauses. This ends up with a very inefficient multiplication of the invocations. We also tested another version of $Q_1$ that consists of a union of the four SERVICE clauses rather than the conjunction thereof. This query should return the same results since all four SERVICE clauses are independent. This however did not change Virtuoso's behavior that kept generating a useless multiplication of the invocations.

**Listing 5.** Query $Q_1$ uses four SPARQL micro-services to retrieve data related to species *Delphinus delphis* from the Biodiversity Heritage Library, Flickr, the Macaulay Library, and MusicBrainz.

```
PREFIX s: <http://schema.org/>
CONSTRUCT {
  <#>
    s:mainEntityOfPage ?article;   # Biodiversity Heritage Library
    s:image ?photo;                # Flickr
    s:audio ?audio;                # Macaulay Library
    s:subjectOf ?mbzPage.          # MusicBrainz
} WHERE {
    SERVICE <http://example.org/bhl/getArticlesByTaxon?name=Delphinus+delphis>
    { ?article s:name ?articleTitle; s:author ?articleAuthorName. }

    SERVICE <http://example.org/flickr/getPhotosByGroupByTag? \
            group_id=806927@N20&tags=taxonomy:binomial=Delphinus+delphis>
    { ?photo s:contentUrl ?img. }

    SERVICE <http://example.org/macaulaylibrary/getAudioByTaxon? \
            name=Delphinus+delphis>
    { ?audio s:contentUrl ?audioUrl. }

    SERVICE <http://example.org/musicbrainz/getSongByName? \
            name=Delphinus+delphis>
    { [] s:sameAs ?mbzPage; s:name ?mbzTitle. }
}
```

**Listing 6.** Query $Q_2$. Invocation of a SPARQL micro-service using a variable endpoint URL: Variables ?bhl, ?flickr, ?macaulay, and ?mzbrnz are built dynamically using the species name in variable ?species.

```
PREFIX s: <http://schema.org/>
CONSTRUCT {
  <#>
    s:mainEntityOfPage ?article;   # Biodiversity Heritage Library
    s:image ?photo;                # Flickr
    s:audio ?audio;                # Macaulay Library
    s:subjectOf ?mbzPage.          # MusicBrainz
} WHERE {
    BIND("Delphinus+delphis" as ?species)

    BIND(IRI(concat("https://example.org/bhl/getArticlesByTaxon? \
        name=", encode_for_uri(?species))) as ?bhl)

    BIND(IRI(concat("https://example.org/flickr/getPhotosByGroupByTag \
        ?group_id=806927@N20&tags=taxonomy:binomial=", \
        encode_for_uri(?species))) as ?flickr)

    BIND(IRI(concat("https://example.org/macaulaylibrary/getAudioByTaxon? \
        name=", encode_for_uri(?species))) as ?macaulay)

    BIND(IRI(concat("https://example.org/musicbrainz/getSongByName? \
        name=", encode_for_uri(?species))) as ?mzbrnz)

    SERVICE ?bhl { ?article s:name ?articleTitle; s:author ?author. }
    SERVICE ?flickr { ?photo s:contentUrl ?img. }
    SERVICE ?macaulay { ?audio s:contentUrl ?audioUrl. }
    SERVICE ?mzbrnz { [] s:sameAs ?mbzPage; s:name ?mbzTitle. }
}
```

**Table 3.** Execution time (in seconds) of query $Q_1$ on Corese-KGRAM and Virtuoso OS Edition, without or with cache database.

| SPARQL Engine | Exec. Time without Cache | Exec. Time with Cache |
|---|---|---|
| Corese-KGRAM | $3.66 \pm 0.10$ | $1.68 \pm 0.04$ |
| Virtuoso | $411 \pm 1$ | $90.3 \pm 1$ |

The evaluation of $Q_1$ by Corese-KGRAM took an average 3.66 s without using the cache database, while the four individual invocations sum up to 2.59 s (2nd column of Table 2). Similarly, the evaluation took an average 1.68 s when using the cache while the individual invocations sum up to 1.16 s (3rd column in Table 2). The overhead of 1.07 s and 0.52 s respectively can be attributed to the query processing by Corese-KGRAM's SPARQL engine as well as networks overheads. The log also revealed that the SERVICE clauses are evaluated sequentially, although they could be evaluated in parallel since they are independent from each other. We discussed this question with the developers who confirmed that, as of today, Corese-KGRAM does not support parallel evaluation of independent SERVICE clauses.

## 6. Biodiversity-Related Use Cases

### 6.1. Aggregating Various Types of Data Related to Biological Taxa

Many existing data aggregators in the biodiversity domain consolidate data with respect to a certain perspective (species occurrences, life traits, taxonomy etc.) from multiple sources. Yet, orthogonal approaches lack that which would allow biologists to aggregate and visualize data spanning these different perspectives. To address this need, we have designed an application that provides biologists with a convenient way to get a quick outlook at the various types of data collected by independent biodiversity programs about a given taxon.

In a joint initiative with the French National Museum of Natural History, we have produced a dataset called TAXREF-LD [29], a Linked Data representation of TAXREF [30], the French taxonomic register for fauna, flora, and fungus (https://inpn.mnhn.fr/programme/referentiel-taxonomique-taxref?lg=en). It models 236.000+ biological taxa along with the 509.000+ scientific names used to refer to the taxa. TAXREF-LD is accessible through a public SPARQL endpoint (TAXREF-LD public SPARQL endpoint: http://taxref.mnhn.fr/sparql), and the URIs of all taxa and scientific names are dereferenceable.

In the application we present here, a SPARQL CONSTRUCT query (the full query is available at https://github.com/frmichel/sparql-micro-service/tree/0.1.0/demo/query/query.rq) first retrieves from TAXREF-LD the information available about the taxon that bears a given scientific name. Then, it enriches TAXREF-LD's taxon description with data from five SPARQL micro-services: BHL, Flickr, Macaulay Library, and MusicBrainz already described in Section 5, and the additional *eol/getTraitsByTaxon* service that retrieves life traits related to a taxon's scientific name from the Encyclopedia of Life trait bank [31].

The RDF graph resulting from this query exhibits URIs identifying each BHL article, each Flickr photo and each Macaulay audio recording. These URIs are built upon the Web API's internal identifier. They are made dereferenceable according to the mechanism we described in Section 4.2: While the micro-services invoked in the SPARQL query retrieve information by scientific name, complementary micro-services retrieve information by Web API's identifier. To do the wiring, the Web server is configured to rewrite a look up query on one of those URIs into the relevant micro-service invocation.

In a next step, the result RDF graph is transformed into an HTML page using the SPARQL Template Transformation Language (STTL) [32]. For each resource retrieved by SPARQL micro-services (e.g., a photo, an article), a SPARQL Template query fetches data from the result graph and passes it on to an HTML template tailored for the resource type. In turn, the latter produces the piece of HTML code necessary to properly display the resource (the code of the STTL transformations is available on

GitHub under https://github.com/frmichel/sparql-micro-service/tree/0.1.0/demo). For instance, a photo is rendered as one element of a carousel while an audio recording is rendered using an HTML5 audio player. A global template assembles the partial HTML snippets to form the final HTML page. Figure 7 depicts a part of the HTML page generated for species *Delphinapterus leucas*.

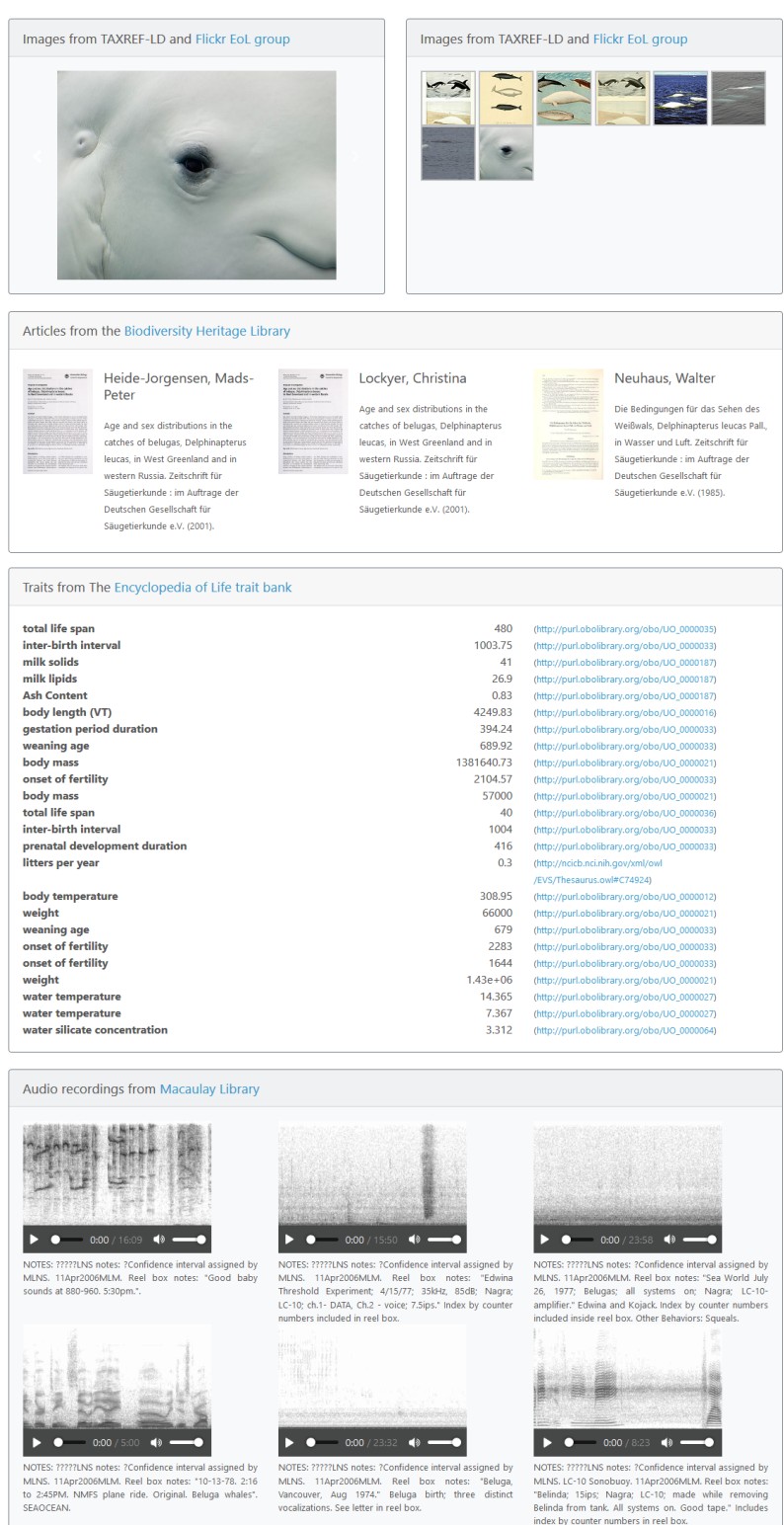

**Figure 7.** Partial HTML rendering of the RDF graph describing the beluga (*Delphinapterus leucas*), constructed by querying Linked Data and multiple SPARQL micro-services. Available at http://sms.i3s.unice.fr/demo-sms?param=Delphinapterus+leucas.

*6.2. Assisting Biologists in Editing Taxonomic Information*

Taxonomic registers are essential tools to comprehend the diversity of nature and develop natural heritage conservation strategies. They are used as the backbone of biodiversity programs and aggregators such as the Global Biodiversity Information Facility (http://gbif.org/) (GBIF) and the Encyclopedia of Life (http://eol.org/) (EoL) that aggregate over 50 data sources each. Unfortunately, there does not exist a single register of the taxonomic knowledge that would mark global consensus. Instead, multiple taxonomic registers cover complementary and often overlapping regions, epochs or domains, while possibly disagreeing on the circumscription of some taxonomic concepts. It is therefore of utmost importance for biologists to be able to confront the knowledge they have about a taxon with related data coming from the manifold data sources available in the biodiversity area.

To tackle this issue, the French Museum of Natural History has developed the TAXREF-Web application (https://taxref.mnhn.fr/taxref-web/) enabling biologists to edit biological taxa's information available in TAXREF. The application offers forms not only to edit TAXREF's database, but more importantly to display data collected from other programs, compare it with TAXREF's data and provide biologists with the ability to add, remove or amend data accordingly. To do so, TAXREF-Web's developers have to write a specific piece of code to query each Web API of interest and manually align the data elements within its results with the corresponding data elements within the results of the other Web APIs. For instance, TAXREF's Web API codes the habitat wherein a species lives as an integer (e.g., 1 stands for marine environment, 2 for fresh water). By contrast, the World Register of Marine Species (http://www.marinespecies.org) (WoRMS) codes the same information as multiple Boolean fields (e.g., isMarine, isFreshwater). Therefore, TAXREF-Web's code must carry out a cumbersome alignment to convert from one representation into the other. This approach is time-consuming in terms of development and maintenance, and hardly scales to many more Web APIs.

In its current status, the TAXREF-Web application queries the following programs: WoRMS, Fishbase (http://fishbase.org), Index Fungorum (http://www.indexfungorum.org/), Pan-European Species directories Infrastructure (http://www.eu-nomen.eu/pesi/) (PESI), and World Spider Catalogue (https://wsc.nmbe.ch/). Depending on the programs, the Web interface may simply present a link to the program's corresponding Web page, point out a disagreement, or point out a disagreement along with the means for a biologist to update TAXREF accordingly. The considered disagreements pertain to taxonomic information in the following situations:

- Synonymy disagreement: A taxon may be associated with a reference name (the preferred name used to refer to the taxon) and a set of synonyms. A disagreement may occur when a program states a reference name that is considered as a synonym in TAXREF, or when they disagree on the synonyms.
- Taxonomic rank disagreement: A taxon or name has different taxonomic ranks in different programs. For instance, a taxon is considered as a species in TAXREF but as a sub-species in WoRMS.
- Author disagreement: Different author names or different spellings and/or abbreviations.
- Habitat disagreement.

We are currently developing a new implementation of TAXREF-Web based on SPARQL micro-services, covering the set of Web APIs considered so far in TAXREF-Web and extending it with the Web APIs of GBIF, EoL, Zoobank (http://zoobank.org/), the International Plant Names Index (http://www.ipni.org/) (IPNI), Flickr and the Macaulay library. Furthermore, we intend to extend the functional perimeter so as to address the following situations:

- Bibliographic references: Retrieve bibliographic references currently unknown in TAXREF, or point out and fix inconsistent references.
- Life traits: Query WoRMS, Fishbase, and EoL for life traits not referenced in TAXREF, or point out and fix inconsistencies.

- Multimedia material: Display material available in TAXREF, suggest adding links to photos in Flickr and audio/video recordings in the Macaulay library.

Our point is to demonstrate that, once the Web APIs are wrapped in SPARQL micro-services relying on common vocabularies, the integration task is amply simplified: Data aggregation and comparison essentially consist of writing the appropriate SPARQL queries, thus significantly alleviating the development and maintenance efforts.

We now illustrate the use case sketched above in the context of a synonymy disagreement, exemplified in the case of taxon *Acetabularia crenulata*. Let us assume that we have defined SPARQL micro-services for TAXREF and WoRMS Web APIs (TAXREF's Web API allows querying TAXREF's database; it is exploited by the TAXREF-Web application), whose invocations produce the graphs depicted in Listings 7 and 8 respectively. According to TAXREF, *Acetabularia crenulata* is a reference (or "accepted") name, whereas according to WoRMS, it is a synonym name and the reference name should be *Acetabularia (Acicularia) crenulata* instead.

Detecting this disagreement can be achieved easily using SPARQL query $Q_3$ in Listing 9. The query searches solutions where *Acetabularia crenulata* would have status accepted in TAXREF and a different status in WoRMS. If $Q_3$ returns a non-empty solution set, then both sources disagree and the solution provides the reference name according to WoRMS. In turn, the Web application must point out this disagreement and provide the user with a way to accept or dismiss this change. Figure 8 depicts an extract of TAXREF-Web's edition page where this option is denoted by the red button "MAJ REF" (update reference name). A similar SPARQL query (not shown here) can detect a disagreement with respect to the author's name spelling, "J.V.Lamour." in TAXREF and "J.V.Lamouroux" in WoRMS. The red button "MAJ AUTEUR" (update author) allows to acknowledge this change in TAXREF.

Let us finally mention that example query $Q_3$ may be invoked on the Web server side, as part of the HTML page generation process, as well as on the client side using asynchronous queries processed in JavaScript. An advantage of the latter is to alleviate the load required to generate the page on the server's side.

**Listing 7.** RDF graph produced by invoking the TAXREF micro-service at http://example.org/taxref/getNameDetails?name=Acetabularia+crenulata.

```
PREFIX dwc: <http://rs.tdwg.org/dwc/terms/>
PREFIX tn:  <http://rs.tdwg.org/ontology/voc/TaxonName#>

<#> dwc:scientificName "Acetabularia crenulata";
    tn:authorship "J.V.Lamour., 1816";
    tn:rank <http://rs.tdwg.org/ontology/voc/TaxonRank#Species>;
    dwc:taxonomicStatus "accepted".
```

**Listing 8.** RDF graph produced by invoking the WoRMS micro-service at http://example.org/worms/getNameDetails?name=Acetabularia+crenulata.

```
PREFIX dwc: <http://rs.tdwg.org/dwc/terms/>
PREFIX tn:  <http://rs.tdwg.org/ontology/voc/TaxonName#>

<#> dwc:scientificName "Acetabularia crenulata";
    tn:authorship "J.V. Lamouroux, 1816";
    tn:rank <http://rs.tdwg.org/ontology/voc/TaxonRank#Subspecies>;
    dwc:taxonomicStatus "unaccepted";
    dwc:acceptedNameUsage "Acetabularia (Acicularia) crenulata".
```

**Listing 9.** Query $Q_3$ detects the disagreement between TAXREF and WoRMS with respect to the status (reference vs. synonym) of *Acetabularia crenulata*.

```
PREFIX dwc: <http://rs.tdwg.org/dwc/terms/>

SELECT ("Acetabularia crenulata" as ?taxrefRefName) ?wormsRefName WHERE {

  SERVICE <http://example.org/taxref/getNameDetails?name=Acetabularia+crenulata>
  { [] dwc:taxonomicStatus "accepted". }

  SERVICE <http://example.org/worms/getNameDetails?name=Acetabularia+crenulata>
  { [] dwc:taxonomicStatus ?s2;
      dwc:acceptedNameUsage ?wormsRefName.
    FILTER ?s2 != "accepted"
  }
}
```

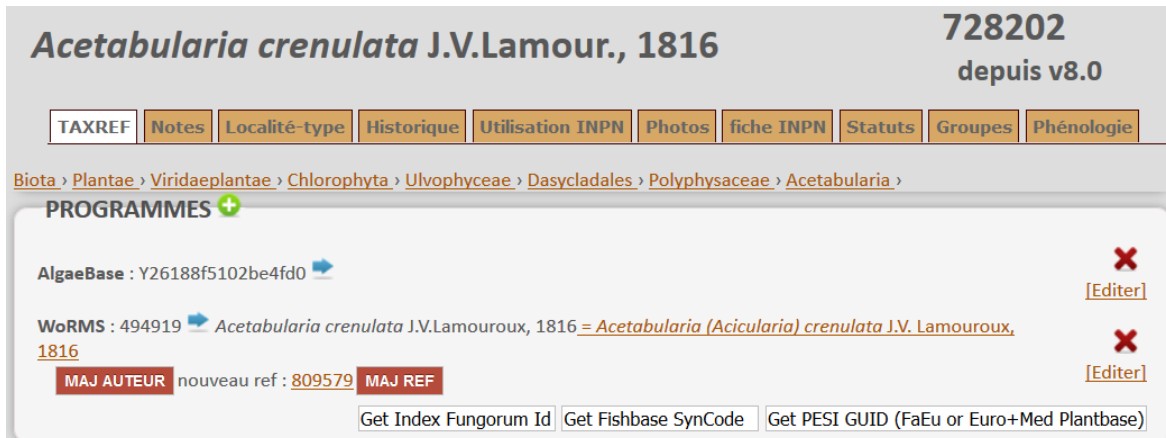

**Figure 8.** TAXREF-Web suggestions for editing taxon *Acetabularia crenulata*.

## 7. Related Works

Integrating heterogeneous data sources has been the object of abundant literature since the early 1990's [33]. Traditionally, data source wrappers carry out the mediation between multiple specific data schemas and a target schema (possibly represented by means of an ontology). To process user queries, a query federation engine determines candidate data sources (the ones likely to be relevant for the query), establishes a query plan, queries each data source accordingly, and recombines the partial results. Our approach proposes a lightweight method to wrap Web APIs and equip them with a SPARQL interface. Although the federation of such wrappers is out of the scope of this paper, existing federated query engines could be adapted to rewrite parts of a client's SPARQL query into SERVICE clauses querying SPARQL micro-services and regular SPARQL endpoints alike.

Translating heterogeneous data sources into RDF triples has been a preoccupation since the early days of the Semantic Web (see the list hosted on W3C's Web site: https://www.w3.org/wiki/ConverterToRdf). Still, approaches specifically concerned with Web APIs are often ad hoc solutions. For instance, Flickcurl (http://librdf.org/flickcurl/) is a hardwired wrapper for Flickr's Web API services. Twarql [34] wraps Twitter's Web API to enable filtering and analysis of streaming tweets. It encodes tweets content in RDF using common vocabularies and enables SPARQL querying. Yet, this approach is very specific to the Twitter and micropost content in general.

With a similar rationale, Hydra [27] is a vocabulary aimed to describe Web APIs in a machine-readable format. It puts a specific focus on the generation of hypermedia controls so as to enable the generation of truly RESTful interfaces. Hydra, used in conjunction with JSON-LD, forms a basis to build hypermedia-driven Linked Data interfaces [35]. This basis can be harnessed to

turn existing Web APIs into RESTful Linked Data interfaces whose documentation can be interpreted at run time by a generic Hydra client. Our incentive with SPARQL micro-services is to provide client applications with the whole expressiveness of the SPARQL Query language, which would be more difficult to achieve using a Hydra-described REST interface.

Closer to our work, Linked REST APIs [36] is a framework dedicated to the semantic annotation of Web APIs and the automatic specification of SPARQL query execution plans that invoke these Web APIs. A major difference with our approach is that the framework requires the deployment of a middleware that stores the Web APIs descriptions and their mapping towards domain ontologies, and computes and enacts query execution plans. The SPARQL micro-services architecture, on the other hand, relies on a totally distributed architecture wherein independent service providers may publish SPARQL micro-services usable by regular SPARQL clients. Yet, a key point for the SPARQL micro-services architecture to scale up easily will be to address the description of micro-services. In the last section, we suggest several leads in this respect.

SPARQL-Generate [37] extends SPARQL 1.1 to enable querying RDF graphs along with non-RDF documents. A SPARQL-Generate query relies on several extension functions to fetch and parse documents in different data formats, and defines the shape of RDF triples to be produced thenceforward. As such, it could be used to query a Web API in a way similar to that of a SPARQL micro-service. Two main differences can be observed though. (i) SPARQL-Generate is an extension of SPARQL, hence, by definition, it is not supported by engines strictly complying with the SPARQL Query language. By contrast, our vision is that multiple service providers could publish independent SPARQL micro-services, thereby building up an ecosystem of services all complying with standard SPARQL; (ii) SPARQL-Generate offers the advantage that querying remote data sources is performed within a single language. On the one hand, this only requires skills with Semantic Web technologies. On the other hand, this entails that a significant part of the whole process is left to the SPARQL client: Querying the data source while providing necessary arguments, and translating its proprietary vocabulary into RDF triples aligned on common vocabularies. Consequently, as illustrated by authors' examples, the additional syntactic sugar required can make queries considerably cumbersome and difficult to maintain. We take a different option where this complexity is hidden from the client and handled by the SPARQL micro-service developer.

An approach very similar to SPARQL-Generate is proposed in [38]. It is based on the BIND_API clause, an extension of the SPARQL BIND clause, that binds a set of variables with values extracted from a Web API response. It suffers the same pitfalls as SPARQL-Generate with respect to our goals: The use of non standard SPARQL Query Language and the cumbersome syntactic sugar left to the SPARQL client.

ODMTP [39], On-Demand Mapping using Triple Patterns, is an attempt to query non-RDF datasets as Triple Pattern Fragments. The authors have implemented a prototype to query Twitter's Web API, that can process triple pattern queries over the whole Twitter's dataset. Conversely, SPARQL micro-services support arbitrary SPARQL queries over restricted fragments of the Web API dataset. Besides, unlike SPARQL micro-services, ODMTP cannot assign dereferenceable URIs to Web API resources. Nevertheless, ODMTP offers the TPF's paging mechanism that SPARQL micro-services should regard as a valuable extension within future works (see the discussion in Section 8).

Our implementation of SPARQL micro-services maps a Web API response to RDF triples in two steps: The response is first translated to JSON-LD, then a SPARQL INSERT or CONSTRUCT query complements the process for cases where JSON-LD is not expressive enough. Alternatively, we could rely on a mapping description language such as RML [40] and xR2RML [41], but they require the developer to learn the mapping language. By contrast, in our proposition we strove to rely only on existing standards.

Let us finally mention the Apache Marmotta project (http://marmotta.apache.org/), a comprehensive Linked Data application framework that implements the Linked Data Platform W3C recommendation [19]. Among others, it provides client modules that wrap the Web APIs of

several Web portals such as Vimeo, Youtube, and Facebook. Hence, it should be relatively easy to implement SPARQL micro-services on top of Marmotta. However, the examples show that the Web API wrapping and the mapping towards RDF triples are mostly hard-coded within the client libraries. Our point is to make the deployment of new SPARQL micro-services as simple as writing a SPARQL query and a configuration file.

## 8. Conclusions and Perspectives

The SPARQL Micro-Services architecture proposes a lightweight type of Linked Data Fragment interface that enables combining Linked Data with data from non-RDF Web APIs. SPARQL querying and URI dereferencing are supported against a virtual graph generated at run-time. This graph is shaped by the Web API service being wrapped, the arguments passed to the SPARQL micro-service and the types of RDF triples that the SPARQL micro-service is meant to spawn. In accordance with the micro-service architecture principles, a SPARQL micro-service should be designed so as to be loosely coupled (it can be deployed independently of other services, possibly using lightweight container technologies), fine-grained (its function is to provide access to a small graph centered on a specific resource such as a photograph, a tweet, or the measure of a sensor). Furthermore, the prototype implementation that we propose is lightweight and simple (configuration-based provisioning, alignment with common/domain vocabularies carried out using a simple SPARQL query).

We think that this approach could promote the emergence of an ecosystem of SPARQL services published by independent service providers, allowing Linked Data-based applications to glean pieces of data from a wealth of distributed, scalable and reliable services. For such an ecosystem to arise however, several crucial issues shall be tackled. Firstly, to enable services discovery, SPARQL micro-services should provide machine-processable self-describing metadata such as the expected arguments, the way they are passed to the micro-service (e.g., as HTTP query string parameters), and the typical graph that may be generated. Secondly, writing SPARQL queries invoking a handful of SPARQL micro-services is easily achieved manually. By contrast, considering a larger number of services should involve the automatic composition of micro-services. The latter may be achieved only if micro-services provide a functional description of the operation they carry out. In this respect, the many works on automatic semantic services composition and multi-agents approaches may be leveraged. Thirdly, although we envision SPARQL micro-services as a way to access small fragments, it should be possible to retrieve such fragments by smaller pieces using a paging mechanism.

To tackle those issues, Verborgh et al. advocated that Linked Data Fragments should provide self-describing, uniform interfaces consisting not only of data triples but also metadata and hypermedia controls. Hypermedia controls contain the information needed to interact further on with a resource. In particular, they allow a client to navigate from one fragment (or a page thereof) to another one [23]. Following up on this idea, the interface of SPARQL micro-services could be extended to return regular SPARQL results alongside additional triples (or quads more generally) representing metadata and control information. In the case of CONSTRUCT and DESCRIBE queries, metadata and controls would be provided as additional triples possibly embedded within a dedicated graph, as proposed in the Triple Pattern Fragments. The case of ASK and SELECT queries may seem less obvious since these do not return triples nor quads but variable bindings. Yet, the SPARQL Results specifications (in XML [42] and JSON [25]), mention the optional *link* header meant to "to refer for further information". This header could be used to link the results to the URI of a graph or a document providing metadata and control triples separately. We can think of this solution as some sort of *Graph Pattern Fragment* interface, i.e., a generalized TPF interface that accepts regular graph patterns instead of only triple patterns, but still complies with the TPF metadata and hypermedia controls specification.

As a final remark, let us underline that this article is focused specifically on consuming Web API data with SPARQL. In a broader picture however, the micro-service architectural principles could be applied to other types of APIs, so as to enable Semantic Web applications to reach out to other data sources, thus participating in the emergence of a more decentralized Web [43]. Furthermore,

many APIs provide read-write access, empowering users to interact with contents, other users, etc. Hence, an interesting perspective would be to think of SPARQL micro-services as a way to support distributed SPARQL Update over Web APIs, thus eventually contributing to build an actual read-write Web of Data.

**Author Contributions:** Writing—original draft, F.M.; Writing—review & editing, F.M., C.F.Z., O.G. and F.G.

**Funding:** This research received no external funding.

**Conflicts of Interest:** The authors declare no conflict of interest.

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
