# Peer review of "Integration of Web APIs and Linked Data Using SPARQL Micro-Services—Application to Biodiversity Use Cases"

_information, doi:10.3390/info9120310_

Reviewer 1 Report

This paper presents a method to integrate data services available from RESTful APIs as ontology data through SPARQL query interface. Using this method, RDF graphs are constructed from data retrieved from Web service invocations. The RDF graph construction is specified with query statements in combination with Web service calls. Authors described implementation as well as some experiments that show modest runtime overhead.

Comment:

In general, the paper is well-written and easy to read. The motivation is convincing and the method description is sufficiently detailed but the presentation may be improved a little.

I have a suggestion for the authors:

Have a section for the formal definition of your design.

The informal discussion in Section 3 is rather unclear. You may want to move the discussion of caching to the implementation section. You may also want to delay the example after you have given a complete definition of the design. Section 3.2 seems belong to discussion or related work. 

You may want to merge part of Section 3 with Section 4 to give a more coherent discussion of your methodology first. 

page 3, line 91, leveragning --> leveraging

Author Response

Dear reviewer,

we wish to thank you for your remarks and advices. Here is a description of how we took them into account.

> Have a section for the formal definition of your design.
> The informal discussion in Section 3 is rather unclear. You may want to move the discussion of caching to the implementation section. You may also want to delay the example after you have given a complete definition of the design. Section 3.2 seems belong to discussion or related work.
> You may want to merge part of Section 3 with Section 4 to give a more coherent discussion of your methodology first.

Thank you for these comments, these are really helpful indeed. We have reorganized section 3.1 to make it more focused on the architectural principles and less intertwined with technical considerations. In particular, we moved the discussion on caching strategy to the section on the implementation.

We have moved section 4, about URI dereferencing, to section 3.2. This way, section 3 is a more natural presentation of the principles and how they should be used to realize a coherent ecosystem of SPARQL micro-services and dereferenceable URIs. We have considered moving the examples into a single dedicated section, but we felt that it was making understanding more difficult. We opted for putting them at the end of each section 3.1 and 3.2, making reading smoother, we think.

The discussion about the Linked Data Fragments now comes as the final part of section 3. We feel that it is useful to keep it in this section rather than in related works, because it does not just compare the proposed approach to other ones, it also gives keys to understand where our approach lies within a whole spectrum of Linked Data interfaces. Please let us know if you would still prefer to move the discussion to the related works section.

Franck MICHEL.

Reviewer 2 Report

The manuscript presents an interesting approach for providing SPARQL access and deferenceable URIs for datasets published as web APIs. The system is based on custom, API-specific wrapper implementations. The motivation for this is to facilitate interoperability between different web APIs, by using shared vocabularies and standard technologies, instead of ad hoc method names, parameters and result formats. This would allow users to query and combine the results of multiple APIs in one query.

Concerning the passing of the API arguments as HTTP query strings to the SPARQL micro-service endpoint: as the authors themselves point out, passing the arguments as SPARQL query triple patterns (instead of HTTP query strings) would feel compelling. This would make writing queries such "get the Flickr photos tagged with a name of a species that belongs to a genus X" easier and more intuitive - and such queries might not even be possible in all SPARQL engines, e.g. Virtuoso, I believe, as variable endpoint URLs in SERVICE are not supported. That way, all the species belonging to a genus X could be computed first in a SPARQL sub-query using a taxonomic dataset's endpoint (SERVICE), e.g. TAXREF, and then the species names bound to a variable could be used in the query for getting the photos of individual species. On other hand, what would be the contents of the default graph of the SPARQL endpoint (http://example.org/flickr/getPhotosByGroupByTag) in such a case?. Intuitively, the default graph would contain all the photos in Flickr, but making a web API call to get that data might be impossible (or in services with less data, at least extremely slow). So considering that, the default graph being a small, resource-centric virtual graph of a single photo makes sense. This of course is dependent of the individual API, as some APIs may return, e.g. all the photos through a single API call (e.g. getAllPhotos).

Regarding Virtuoso's query performance (discussed in section 6.2) - would it help if UNION would be used to combine the different SERVICEs?

Minor/language remarks:

Line 276: "be achieved" -> "is achieved"?

Line 367, 449, and footnote 21: "Github" -> "GitHub"

Line 368: "Docker hub" -> "Docker Hub"

Line 456 and 545: "macaulaylibrary/getAudioById"  vs. in Listing 5 and 6 you use "macaulaylibrary/getAudioByTaxon"

Line 470: "CONSTRUCT WHERE ?s ?p ?o" -> "CONSTRUCT {?s ?p ?o} WHERE {?s ?p ?o}"

Figure 5 and 6: "Macauley" -> "Macaulay"

Author Response

Dear reviewer,

we wish to thank you for your remarks and advices. Here is a description of how we took them into account.

Discussion with respect to how arguments are passed to a SPARQL micro-service.

We definitely agree about the compelling aspect of passing arguments in the graph pattern, and the fact that it makes queries more natural. This however comes with certain concerns that we shortly point out. To justify our choice, we have slightly extended this discussion, mentioning that providing the arguments as terms of the SPARQL graph pattern requires a mechanism to identify these specific terms as the service inputs. This actually opens the door to a much larger discussion about the description of semantic Web services, their (automatic) discovery and composition. We just touch upon this in the conclusion, but we feel like this discussion goes beyond the scope of this article. Note that we are currently working on extending SPARQL micro-services towards this direction.

> Line 456 and 545: "macaulaylibrary/getAudioById"  vs. in Listing 5 and 6 you use "macaulaylibrary/getAudioByTaxon"

Indeed, we used the getAudioById service for comparing measures in section 6, and getAudioByTaxon to retrieve recordings by taxon name in the use case (section 7.1).

Behind the scene, getAudioByTaxon first calls a Web API service to get the id corresponding to the name, and in turn the service to get recordings for that id. Consequently, we did not want to use getAudioByTaxon to assess performances because it invokes the Web API twice. Instead, we chose to use getAudioById that invokes the Web API only once, making it comparable to other services.

> Regarding Virtuoso's query performance (discussed in section 6.2) - would it help if UNION would be used to combine the different SERVICEs?

Very interesting suggestion. The answer is no: even with a union  of all SERVICE clauses, each one is invoked many times, once for each solution computed from the SERVICE clauses previously evaluated, leading to an “invocation product”. We added a sentence about this suggestion in section 6.2

Best regards,

   Franck MICHEL.

Round  2

Reviewer 1 Report

I read the updated version and see that the authors have reorganized the sections as I suggested. I think the present form is good for publication.

Author Response

Dear reviewer,

Thank four for this final review.

Best regards,

   The authors.

Reviewer 2 Report

The authors have taken all my previous comments into consideration, extended the weak sections of the manuscript, and thereby improved the quality of the paper.

Once the minor revisions stated below are taken into account I think the manuscript is ready for publication.

Minor/language remarks:

1)

>> Line 456 and 545: "macaulaylibrary/getAudioById" vs. in Listing 5 and 6 you use "macaulaylibrary/getAudioByTaxon"

> Indeed, we used the getAudioById service for comparing measures in section 6, and getAudioByTaxon to retrieve recordings by taxon name in the use case (section 7.1).

> Behind the scene, getAudioByTaxon first calls a Web API service to get the id corresponding to the name, and in turn the service to get recordings for that id. Consequently, we did not want to use getAudioByTaxon to assess performances because it invokes the Web API twice. Instead, we chose to use getAudioById that invokes the Web API only once, making it comparable to other services.

That makes sense, but if I understood your presentation correctly, the queries in Listing 5 and 6 are the ones used for comparing measures, where you should be using getAudioById. Instead, in the Listing 5 and 6 you have getAudioByTaxon. Should it be changed to getAudioById?

2)

Section numbering: there are two Section 3's:

 3. The SPARQL Micro-Service Architecture

 3. Implementation

The latter one should be Section 4.

3)

Line 512: "%DIF >" - If this is part of the text (and not just some technical diff notation), it should be removed.

Author Response

Dear reviewer, thank you for these final remarks.

> That makes sense, but if I understood your presentation correctly, the queries in Listing 5 and 6 are the ones used for comparing measures, where you should be using getAudioById. Instead, in the Listing 5 and 6 you have getAudioByTaxon. Should it be changed to getAudioById?

My mistake. Actually the queries in listings 5 and 6 are the right ones, but I forgot to introduce service getAudioByTaxon in section 5.2 instead section 6. I updated both sections.

> 2) Section numbering: there are two Section 3's

> 3) Line 512: "%DIF >" - If this is part of the text (and not just some technical diff notation), it should be removed.

Actually I cannot see those issues. Perhaps they only appear in the PDF that the editor regenerated from the Latex files. I will double-check that with them.

Thank you for your help and advices.